equity; digital health technologies; intersectionality; access and inclusion; virtual reality

**Corresponding author:**
Martin Lepage;
Email: Martin.lepage@mcgill.ca

# Virtual reality offerings for wellbeing for and by marginalized populations: A scoping review on equity and intersectionality

Quinta Seon[1,2] , Adèle Hotte-Meunier[2,3], Lisa Sarraf[2], Caroline Dakoure[2], Clayton Jeffrey[1,2], Ethan C Draper[4,5], Geneviève Sauvé[1,2,3], Myrna Lashley[1] and Martin Lepage[1,2]

[1]Department of Psychiatry, McGill University, Montréal, QC, Canada; [2]Research Centre, Douglas Mental Health University Institute, Verdun, QC, Canada; [3]Department of Education and Pedagogy, Université de Québec à Montréal, Montréal, QC, Canada; [4]Department of Neuroscience, McGill University, Montréal, QC, Canada and [5]Montreal Neurological Institute, McGill University Health Centre, Montréal, QC, Canada

## Abstract

Although virtual reality (VR) programs are being developed by marginalized groups', a systemic power imbalance still exists. Marginalized groups have a place in digital wellbeing and can lead initiatives to access resources that they desire. To better support these efforts and mobilize knowledge among marginalized stakeholders, we conducted a scoping review of the use of VR for wellbeing. Adopting an equity lens that considers the experiences of intersectional marginalization, our aim was to identify VR programs, their targets, outcomes and equity-related facilitators and barriers. In May 2023, we conducted a comprehensive literature search of MEDLINE, PsycINFO, Embase and Web of Science databases and grey literature for *virtual reality* and *marginalized populations*. Eligible research articles since the inception of the databases were those that met our predefined criteria of VR, marginalized populations and wellbeing. We included 38 studies and charted preregistered variables using narrative synthesis, descriptive statistics, and a logic model. The populations were often intersectionally marginalized–primarily individuals with disabilities, underrepresented sexualities and genders, and marginalized older individuals in high-income countries on Turtle Island (North America). The most common race categories were Black or African American (26%) and European or White (53%), but other sociodemographic characteristics were underreported. VR offered diverse support, including social, mental, physical and cultural. We report program outcomes for several subgroups; though heterogeneous, most studies reported improved wellbeing outcomes. VR's flexibility created informal, flexible spaces, with peer support that contributed to mental and social wellbeing. Several factors could hinder marginalized groups' ability to access and participate, such as the lack of free programs, data and program ownership, and intersectional data analyses. This topic reflects a growing literature, with half of the publications being in 2022 or 2023. Many of these studies have limitations like small sample sizes and a lack of mixed-methods or practical significance analyses. Moving forward, researchers should apply more open-access and inclusive practices in their designs and recruitment processes to widen equitable access to marginalized stakeholders. Nevertheless, many marginalized populations created VR programs and benefited from them, contributing to a rebalancing of power over wellbeing.

## Impact statement

The variety of uses of VR, and the fact that it can be customized, means it has the potential to be further utilized by marginalized communities. To share knowledge with marginalized stakeholders, we reviewed research on VR's use for wellbeing. We also assessed equity- and intersectionality-related factors. We found that VR has broad benefits and uses, meeting users where they were at, but that some questions remain. How can VR programs be owned by marginalized populations and continued beyond just research projects? How can intersectionality, the impact of multiple oppressions, be better included in the design and research on this topic? Could further community engagement help identify relevant targets and mechanisms outside of the mainstream? We discuss these challenges and provide recommendations. We gather information and potential resources for marginalized clinicians, researchers and populations to use VR programs and factors that facilitate their use.

## Introduction

### Background

Virtual reality (VR) could improve the wellbeing of marginalized groups if it is implemented equitably (World Health Organization, 2008; Social Determinants of Health Framework Task Group, 2015; Wassell and Dodge, 2015). "Wellbeing" refers to the composite of challenges and strengths experienced by a person, such as socioeconomic factors or mental and physical health. VR is an immersive, three-dimensional experience with which users can interact. Through a head-mounted device or on a computer, VR communicates an experience of an alternate reality, which the user can influence (Sherman and Craig, 2018). A technology with a slew of applications for wellbeing, VR can induce relaxation, encourage health behaviors, help with mental health rehabilitation, educate, enhance social support and more (Liu et al., 2022; Singha and Singha, 2024). This article reviews the literature on VR programs designed to support the wellbeing of marginalized groups.

To date, VR has not been inclusive. When people are excluded from resources and exposed to wellbeing risks, i.e., marginalized, they are being oppressed by multiple systems working together (Crenshaw, 1989; Robards et al., 2018; Baah et al., 2019; Schueller et al., 2019; Rami et al., 2023). In this text, we use the verbs "marginalized" and "marginalization" to refer to the unfair conditions imposed on people. Marginalization is a societal process, not an identity, and various inequities sustain it (Wingrove-Haugland and McLeod, 2021; Fluit et al., 2024). For instance, poorer wellbeing outcomes are strongly linked with experiences of racism, heterosexism and other marginalizations (Paradies, 2016; Plöderl and Tremblay, 2015). Similarly, the lack of inclusivity and representation of diverse experiences in digital health is marginalizing. *Intersectionality theory* posits that the mechanisms of marginalization are inextricably linked – dependent on each other to continue this power imbalance (Crenshaw, 1989). Equity provides an opportunity to address the unfair differences between groups regardless of social, political, financial or other factors (Whitehead, 1992). Improving the lack of equity and intersectionality of VR will thus improve its application in healthcare.

According to a review by Schueller et al. (2019) in the United States, VR development faces access and participation barriers. These barriers are illustrated by a lack of access to technology and a lack of intersectionality and variety in programming. For example, Indigenous participants from nations across multiple high-income Western countries indicated that digital health does not consider their needs related to their age, gender, culture and norms (Li and Brar, 2022; Henson et al., 2023; Hicks et al., 2024). Similarly, digital health solutions have failed to cater to individuals from underrepresented sexualities and genders who desire holistic, interpersonal and positive support (Steinke et al., 2017; Gilbey et al., 2020). Ramos et al. (2022) reviewed diversity, equity and inclusion in mental health mobile applications, and they concluded that demographic information and sample diversity are often underreported in this type of study (Ramos et al., 2021). Intersectionality in this area of research has also been identified as insufficient (Figueroa et al., 2021; Huang et al., 2020). These issues offer marginalized groups a lower standard of care, amplifying healthcare inequities (Whitehead et al., 2023). It creates a perception among users of lack of trustworthiness, safety and irrelevance (Pendse et al., 2022; Whitehead et al., 2023). Thus, the experiences within and across these marginalized groups are varied, making clear that intersectionality becomes a key consideration for VR creators.

Members of marginalized groups are developing VR, which has proven effective in several contexts. Pendse et al. (2022) discussed how digital health programs like VR could incorporate cultural relevance, thereby promoting agency and informal help-seeking in multiple forms. Progress is being made toward these goals, with several proof-of-concept studies being underway for cultural adaptation to key underserved groups (Seon et al., 2023; Trueba et al., 2024). Studies have demonstrated that VR can facilitate experiences, such as cultural or gender affirmation, by enabling users to embody avatars and interact within realistic virtual worlds (Wallis and Ross, 2021; Chong et al., 2022; Dincelli and Yayla, 2022; Freeman and Acena, 2022). Additionally, Second Life, a collection of virtual environments, hosts community-organized gatherings, which have been shown to benefit the mental and social wellbeing of disabled and 2SLGBTQIA+ persons (Acena and Freeman, 2021; Freeman and Acena, 2022). These applications of art, heritage, games and therapy demonstrate how VR can enhance the wellbeing of marginalized groups.

### Objectives

Our aim was to describe the outcomes of these programs while taking into account the experiences of intersectional marginalization. We sought to identify gaps across several variables of interest, such as program targets, equity, access and inclusion. These are defined as conditions affecting wellbeing; actions taken to improve unfair practices; elements of the digital divide, such as financial barriers to technology; and the representation of diverse populations.

### Positionality

The research was developed by a multidisciplinary team of diverse backgrounds living and studying on Turtle Island (North America). We drew from key theoretical literature sources from marginalized authors, additionally. As a diverse team with multiple insights and lived experiences, we reflected and discussed how to make our research inclusive and equitable, such as in research planning, choice of variables, assumptions and interpretations. The team is comprised of both marginalized VR users and creators. However, our work may reflect biases and power dynamics, as we are primarily from academic and healthcare-based backgrounds. We are influenced by our experiences in a high-income Western country, but we reflected on our responsibilities to other communities. Part of our efforts is reporting on this study with the SIITHIA intersectionality checklist, making study materials open access and advocating for equity in healthcare in our work and conference presentations. Our research team descend from a mix of heritages, such as mixed (Q.S.), Black and Caribbean (M.L., C.D. and Q.S.), Asian (L.S.) and European (G.S., A.H.M., E.C.D., C.J., M.Le. and Q.S.). Several members are shaped by lived multicultural experience, intersectional discrimination and racialization (Q.S., L.S., C.D. and M.L.) and settling or immigrating to Turtle Island (all). The majority of the team experience marginalization based on gender and sexuality. Several members also experience biases based on their ability.

## Methods

### Protocols and preregistration

The review protocol was preregistered (https://doi.org/10.17605/OSF.IO/K2EGN). The protocol follows the Preferred Reporting Items for Systematic reviews and Meta-Analyses extension for Scoping Reviews (PRISMA-SCR) (Tricco et al., 2018), Joanna

Briggs Institute methodology for scoping reviews (Peters et al., 2015; Peters et al., 2020) and Strengthening the Integration of Intersectionality Theory in Health Inequality Analysis (SIITHIA) checklist (Blair et al., 2022).

### Research aims and design

Due to its broader framing following the participant, context and concept (PCC) structure, our research question, *how VR is used for wellbeing among marginalized groups?*, suited a scoping review methodology. To summarize the research process and outcomes, such as how targets of VR programs lead to wellbeing outcomes, we developed a logic model (W.K. Kellogg Foundation, 2004).

### Search strategy

On May 8 and 9, 2023, we searched OVID (MEDLINE, PsycINFO, Embase), Web of Science databases and the grey literature using keywords in Google Scholar. These keywords were generated from pre-existing literature filters that we edited to fit our topic and from articles identified during prior limited searches (Walsh et al., 2014; Wafford et al., 2018; NIH, 2019; UofA Subject guides, 2023). The two main search themes were *virtual reality* and *marginalized populations* (Supplementary Material). The search terms included healthcare disparities, social disparity, ethnic populations, disabled, minority group, social discrimination, sexism, Indigenous people, LGBTQ, among hundreds of other variations and terms (combined with or) and virtual reality with other relevant terms.

### Selection criteria

The search included quantitative and qualitative studies from any country, publication date, and language. It excluded user and design studies due to the rare presentation of outcome data, but experimental, observational and descriptive studies were included. Peer-reviewed research and dissertations were included, while other reviews, conference abstracts and letters were excluded as they did not contain original research.

### Screening

#### Inclusion criteria

The review included studies whose populations were marginalized, defined as having limited access to health promotion and greater exposure to health risks (Baah et al., 2019). Some examples are socioeconomic, underrepresented races, underrepresented sexualities or genders, and persons at the intersections of these groupings. Studies must have met the definition of VR from Sherman and Craig (2018): a communicated experience, creation of a virtual world, immersion into an alternate reality and the ability of the user to interact with this world. The articles had to investigate wellbeing. We operationalized wellbeing as the multiple factors within which a person could have varying levels of challenge or resources to reflect the United Nations' (UN) Sustainable Development Goals (Social Determinants of Health Framework Task Group, 2015; Wassell and Dodge, 2015; World Health Organization, 2008). Broadly, these categories were socioeconomic, mental and physical health, social, cultural and spiritual factors.

#### Exclusion criteria

Considering VR as an opportunity for creative, pluralistic and holistic wellbeing (Pendse et al., 2022), it was necessary to map the literature on VR with a broad definition of wellbeing, reflecting the UN's Sustainable Development Goals. While various outcomes, including mental or physical health, could be covered by the included papers, they were only included if their objectives were not based solely on diagnosis–symptom paradigms. For example, VR aiming to improve symptoms of anxiety in individuals with anxiety disorders would be excluded. We made this decision because this is a topic of previous reviews (Ionescu et al., 2021; Rowland et al., 2022; Tassinari et al., 2022) and to respect the broad scope of wellbeing. Anti-prejudice articles were excluded because they often did not include marginalized populations and often focused on empathy-building. Articles about vocational training, short-term disability or occupational groups were excluded for relevance. Serious games, projections and gaming systems that did not include the key components of VR as defined in our protocol were also excluded.

### ASReview and Covidence

References were deduplicated and quality-checked in EndNote 20 (The Endnote Team, 2013). We calibrated rater agreement with 100 articles to train the software for automating abstract screening, ASReview. ASReview (version 1.1) is an open-source software with active learning. It has been validated and used by many reviews with a 5% error rate (van de Schoot et al., 2021). We set a stopping criterion of screening 35% of all articles or 150 irrelevant articles in a row, two raters reviewed titles and abstracts (phase 1). Models were switched midway from default to neural networking and Sentence Bidirectional Encoder Representations from Transformers (SBERT), following the procedures suggested by ASReview in phase two of abstract screening (Teijema et al., 2023). The decisions on articles seen by only one reviewer were confirmed in pairs. One independent rater screened unseen articles deprioritized by ASReview for quality checking. From this, one article was included. Full-text screening was carried out using Covidence, performed by two screeners per article, and disagreements were resolved by consensus (Veritas Health Innovation, n.d.). The first author checked reference lists using *citationchaser* and filtered them with ASReview (Haddaway et al., 2021). Consensus was made on any included articles from citation searching with another rater.

### Data extraction, charting and reporting

Raters piloted a data extraction tool with five articles. We extracted the following preregistered variables: bibliographic data, participant characteristics, characteristics of the VR programs, intervention or program information, outcomes, equity/intersectionality, qualitative results, study characteristics and research team characteristics (detailed in study protocol). Some additional variables of interest emerged: *study limitations* and *main finding summary* to assist in the appraisal of findings and quality of studies. Some of our equity- and intersectionality-related variables were age, gender, sex, race, ethnicity, nationality, sexual orientation, romantic orientation, highest educational achievement, occupation and income. For the interventions, these included: accessibility, co-design or community involvement, critical analysis of inclusion and exclusion criteria, language of the intervention, and others. We report the data with narrative summaries and descriptive analyses, including stratified analyses for intersectionality. Qualitative data were processed via content analysis. We provide a flowchart of the screening process (Haddaway et al., 2022).

Throughout the review process, we used a logic model as suggested by published reviews and guidelines (Kneale et al., 2015; W.K. Kellogg Foundation, 2004). The first author developed the preliminary logic model in iterations with the research team. This logic model helped refine the review question and identify variables

of interest. During data charting and synthesis, the logic model informed our interpretations.

We assessed the equity and intersectionality of the included articles using the SIITHIA checklist (Blair et al., 2022). This tool assists in tracking and reporting health inequalities and is guided by eight principles: intersecting categories, multilevel analysis, power, equity, social justice, time and space, diverse knowledge, and reflexivity (Blair et al., 2022). Our assessment of our review using this tool is presented in the Supplementary Material.

## Results

### Overview of literature findings

#### Identified articles and agreement

We identified 12,401 records before deduplication (Figure 1), and a further 1,724 were identified through backward and forward citation searches. Raters reviewed and included 43 records – 38 studies

as some publications use the same data. Rater agreement was high on average and increased, as measured by Gwet's Agreement Coefficient 1 (AC1)) in phase I (AC1 = .80) and phase II (AC1 = .89) of title and abstract screening and full-text selection (AC1 = .90).

#### Selected literature

The selected literature is listed in Table 1. Most of the literature was from Western or high-income countries (89%) – 55% on Turtle Island – and was published in 2022 or 2023 (45%). The populations experienced marginalization based on disability or developmental disorders (k = 22 studies), underrepresented sexualities and genders (k = 16 studies), age (k = 13 studies), social, economic or political circumstances (k = 5), geographic location (k = 1), race and/or ethnicity (k = 4). Several studies explicitly recruited based on the intersection of these marginalizations (k = 21). There were mental (k = 33), social (k = 15), cultural (k = 2) and physical (k = 10) wellbeing targets.

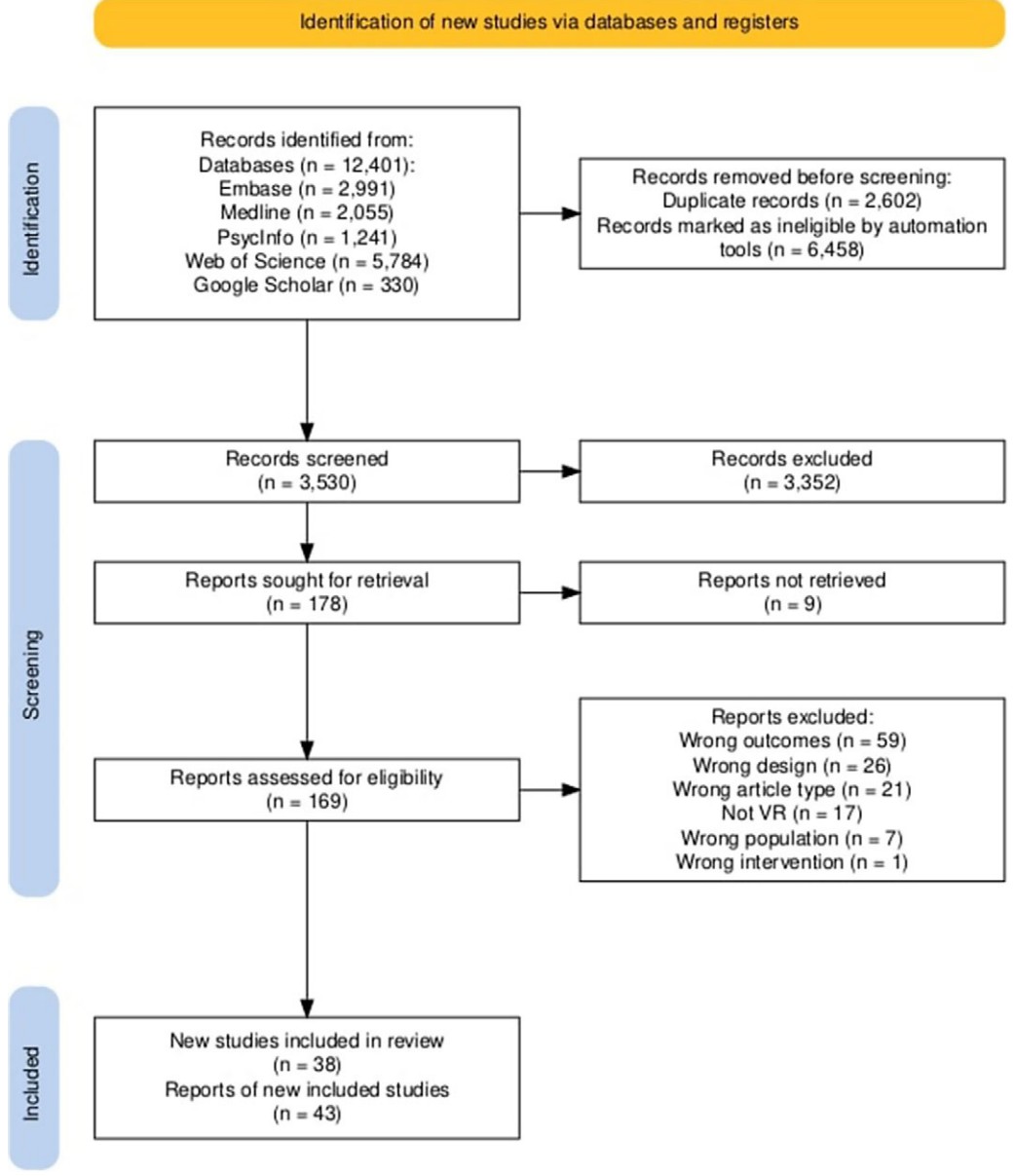

**Figure 1.** PRISMA flowchart.

**Table 1.** Study characteristics

| Studies | Year | Country | Participants | N | Intervention | Aims |
|---|---|---|---|---|---|---|
| *Non-therapy psychological intervention* | | | | | | |
| Shah et al. | 2015 | Singapore | People with mood disorders | 22 | VR-based relaxation and psychoeducation | Investigate feasibility and initial efficacy of the program |
| Nosek et al. | 2016 | Turtle Island (United States) | Women with disabilities | 33 | Self-esteem Second Life | Translate and test the feasibility of the intervention into VR |
| Robinson-Whelen et al. | 2020 | Turtle Island (United States) | Adult women with traumatic spinal cord injury | 12 | VR self-esteem | Investigate feasibility of program adapted for disabilities |
| Wang et al. | 2023 | China | Rural elementary school children from underrepresented backgrounds | 28 | VR social competence education | Investigate the impact of the program on social competence and perceived social support |
| Abal | 2012 | Turtle Island (United States) | Adult English-language learners | 6 | VR language instruction | Investigate intervention's impact on speaking anxiety |
| Chavez et al. | 2020 | Turtle Island (United States) | Youth experiencing homelessness | 10 | VR meditation | Evaluate feasibility of program |
| Lee and Cha | 2021 | South Korea | Female sexual violence survivors (age 18–24) | 19 | VR reflective writing and mindfulness meditation | Describe the feasibility of the intervention and its preliminary effects |
| Health education and support | | | | | | |
| Nosek et al. | 2018 | Turtle Island (United States) | Women with mobility impairments | 24 | VR weight management | Examine differences in weight management outcomes as well as the feasibility of the program |
| Mitchell and Laird; Mitchell et al. | 2022; 2023 | Turtle Island (United States) | Minority women with type 2 diabetes | 158 | Diabetes self-management education and support virtual world | Compare effectiveness of in-person and virtual diabetes groups and explore how the experience of presence in VR could enhance engagement |
| Johnson et al. | 2014 | Turtle Island (United States) | Adults with type 2 diabetes | 20 | Diabetes self-management training | Examine the feasibility and efficacy of the program |
| Rosal et al. | 2014 | Turtle Island (United States) | African American women with type 2 diabetes | 46 | VR diabetes self-management | Evaluate the feasibility and effectiveness of the program |
| Ruggiero et al. | 2014 | Turtle Island (United States) | Low-income African Americans with type 2 diabetes | 41 | Diabetes self-management education virtual world | Examine the acceptance, use and outcomes of the program |
| Wang et al., Christensen et al. | 2021, 2013 | Turtle Island (United States) | Men who have sex with men, aged 18–24 | 444 | Interactive virtual dating narrative game for safe sex | Assess hypotheses that shame and user-avatar bonds account for behavior changes related to program use |
| Bell et al. | 2018 | Turtle Island (United States) | Minority and underserved public school students from third to fifth grade | 116 | Augmented reality health and nutrition education (gardening and cooking game) | Examine program's impact on psychosocial determinants of behavior and dietary intake |
| Cheung et al. | 2022 | China | Individuals with intellectual disability | 42 | VR for life skills (grocery shopping, cooking, kitchen cleaning) | Evaluate the program's impacts on life-skills, self-efficacy, and functioning |
| Leisure and art activities | | | | | | |
| Freeman and Acena, Li et al. | 2022, 2023 | Turtle Island (United States) | Queer social VR users internationally | 29 | Social VR | Examine the experience of embodied visibility and social support in social VR and the impacts on identity practices |
| Chaze et al. | 2022 | Turtle Island (Canada) | Older adults in long-term care homes | 32 | Recreative field trips | Investigate a program to support wellbeing |
| Brimelow et al. | 2022 | Australia (so named) | Cognitively diverse aged care residents living in residential aged care facilities | 25 | In-person social group field trips (leisure) | Assess feasibility of program |
| Brimelow et al. | 2020 | Australia (so named) | Residents of a care home with and without dementia | 13 | Field trips (relaxation theme) | Investigate the effectiveness of the program |
| Afifi et al. | 2022 | Turtle Island (United States) | Family dyads supporting older adults in independent and assisted living | 21 | Virtual field trips and memory sharing with family members | Examine whether the program improves quality of life when family members are at a distance |
| Stendal and Balandin | 2015 | Norway (Sápmi) | One man living with autism spectrum disorder | 1 | Social VR | Explore the VR program and its affordances for autism spectrum disorder |

*(Continued)*

**Table 1.** (*Continued*)

| Studies | Year | Country | Participants | N | Intervention | Aims |
|---|---|---|---|---|---|---|
| Singh et al. | 2017 | Malaysia (so named) | Adults with physical disabilities | 18 | VR sports like tennis, bowling and boxing | Examine the program's impact on psychological wellbeing and upper limb function |
| Reyes and Fischer | 2022 | Turtle Island (United States) | Transgender and genderqueer individuals in VR online communities and forums | 29 | Social VR | Explore self-perception and wellbeing in VR gaming experiences |
| Matsangidou et al. | 2023 | Republic of Cyprus | Middle age to elderly adults with mild dementia or cognitive impairment | 30 | Virtual field trips | Examine elicitation of positive emotional experiences and reduction in negative emotions with VR |
| Lee et al. | 2015 | South Korea | Women over 65 years old | 26 | Individualized feedback-based virtual reality exercise (IFVR) based on tai chi | Examine effect of program on health-related quality of life |
| Acena and Freeman | 2021 | Turtle Island (United States) | LGBTQ gamers ages 18–23 | 8 | Social VR | Report on LGBTQ users' engagement in social VR, especially regarding how social VR may afford social support for these users |
| G. Freeman et al. | 2022 | Missing | VR users who self-identify as non-cisgender | 15 | Social VR | Investigate the strategies non-cisgender users use to build and experience their diverse identities and the challenges these users meet in their identity practices |
| Kwon et al. | 2022 | South Korea | Adults with hemiplegia due to neurological impairments | 21 | VR balance training | Examine the effect of the program on quality of life |
| Bloustien and Wood | 2016 | Australia (so named) | Founders, coordinators and users of Second Life disability-centered spaces | NA | Avatar creation and customization | Explore how VR interacts with self-representation and its potential for social advocacy beyond virtual worlds |
| McKenna et al. | 2022 | Turtle Island (United States) | Transgender and gender diverse adolescents | 10 | Avatar customization | Explore experiences of affirmation and validation during avatar creation |
| Paré et al. | 2019 | Turtle Island (Canada) | Young gender-nonconforming queer game designers | 5 | VR sculpting and VR social environment | Explore representation of gender and sexual orientations with VR |
| Psychotherapy | | | | | | |
| Bond et al. | 2023 | United Kingdom | People with a diagnosis of psychosis | 20 | VR cognitive therapy | Explore user experiences |
| D. Freeman et al., Altunkaya et al., D. Freeman et al. | 2022, 2022, 2022 | United Kingdom | Individuals with schizophrenia spectrum or affective psychosis self-reporting agoraphobic anxiety | 174 | VR therapy (gradual behavioral experiments) | Reduce agoraphobic avoidance of everyday situations and distress when in those situations |
| Beaudoin et al. | 2023 | Turtle Island (Canada) | Adults with treatment-resistant schizophrenia or schizoaffective disorder | 10 | VR for auditory hallucinations | Understand changes in quality of life after treatment |
| Culture and heritage activities | | | | | | |
| Dawson et al. | 2011 | Turtle Island (Canada) | Nine Inuit elders in Nunavut | 9 | 3D reconstruction of Igluryuaq/Thule Whalebone House | Explore how digital replicas of traditional Inuit life can be used in the repatriation of traditional knowledge |
| Park et al. | 2022 | Aoterea (New Zealand) | Female participants identifying as Maori | 5 | Mixed reality cultural heritage site (wharenui) | Show that mixed reality is an effective mechanism for the growing diaspora of Maori to access and experience their language, genealogy, families, histories and knowledge |
| Environmental adaptations | | | | | | |
| Mills et al. | 2023 | Australia (so named) | Adults with disabilities and/or autism spectrum disorder | 31 | VR sensory room | Determine the impact of the program on sensory processing and wellbeing |
| Davis and Chansiri | 2019 | Turtle Island (United States) | Group membership profiles of individuals with disabilities in virtual worlds | 34 | Working in virtual worlds | Understand how visual bias impacts people with disabilities' work experience |

## *Participant demographics*

Overall, the studies included n = 1,587 individuals from various demographic categories (median n = 22 per study [range: 1–444]). The average age was 41.89 [95% CI: 33.05–50.72]. Sex (69% female, no intersex) and gender (71% men) were biased due to the exclusion of non-binary and intersex categories, and the conflation of sex and gender. Non-binary, transgender and gender-non-conforming individuals were underrepresented (at least n = 54 transfemme

and transmascs). Most of the participants were Black/African American (n = 247, 26%) or White/European (n = 502, 53%). Some other race categories were "Other" (n = 44, 5%), Latinx (n = 73, 8%), Mixed (n = 40, 4%), Indigenous (1%) and Asian (2%). Again, there was some conflation of race and ethnicity. When sexuality was not central to the study's aims, it was typically not reported. The highest educational achievement was commonly high school (58% in k = 16 studies). In some studies, the populations were underemployed – about half of those whose occupational status was known in k = 13 studies – or of lower income, but this was reported in very few studies (k = 7). There was an underreporting of carer, stay-at-home and retired occupations overall.

## Study information

The authors reported qualitative (k = 11), mixed methods (k = 6) and quantitative (k = 21) study designs, but almost half of the studies collected some quantitative and qualitative data. The studies used rating scales, interviews, observations, open-ended questions, physiological measures, researcher-developed questions and visual analogue scales. Data collection was typically longitudinal with a median of 8 weeks [IQR: 4–8.5]; the few follow-ups (k = 6) were a median of 4.5 months post-intervention. The quantitative study designs were randomized controlled trials (RCTs) (k = 8), feasibility or pilot (k = 4), quasi-experimental (k = 4), longitudinal or retrospective (k = 3), experimental (k = 1) and case reports (k = 1).

## Narrative summary

We present categories of interventions and their characteristics in Table 2. The program impacts vary based on the population. In the

**Table 2.** Description of interventions

| Type of intervention | k | Intervention contents |
|---|---|---|
| *Non-therapy psychological intervention* | 7 | Interventions based on techniques such as relaxation, meditation, mindfulness or seeking to improve psychological factors without explicit therapeutic techniques. These were for the majority multi-session interventions aiming to have a supportive impact to mental health related variables. |
| *Health education and support* | 8 | Usually gamified or lesson-based activities in virtual worlds or scenarios over multiple sessions. These interventions aimed to improve knowledge, support and self-efficacy on physical health conditions and risks. |
| *Leisure and art activities* | 16 | The impacts of social virtual worlds on psychological wellbeing, social support and identity. Leisure activities in VR included social groups, field trips, creating art and customizing avatars. |
| *Psychotherapy* | 3 | Structured use of therapeutic techniques like cognitive behavioral therapy and avatar therapy for psychosis-related conditions and quality of life. On average, seven sessions in length and targeting fears, avoidance and hallucinations. |
| *Culture and heritage activities* | 2 | These programs promoted a connection to culture and heritage through immersive cultural activities. |
| *Environmental adaptations* | 2 | Virtual worlds can add enabling elements for working and sensory processing. |

next section, we summarize the results by population, including subpopulations for intersectionality considerations. This is followed by a cumulative summary of the literature (Figure 2). Finally, we report on equity and intersectionality.

## Indigenous persons (k = 2)

Cultural heritage was the topic of two qualitative studies delivered to Māori (Aotearoa) (Park et al., 2022) and Inuit elders (Qikiqtaaluk) (Dawson et al., 2011). The programs demonstrated traditional living environments and meeting places – a Thule Whalebone house/ Igluryuq and a wharenui – and passed on knowledge and enjoyable cultural experiences to their communities.

## Marginalized youth (k = 3)

Multi-session, classroom-based programs provided underserved and underrepresented students with guidance from avatars or their teachers on self-efficacy and competence (Bell et al., 2018; Wang et al., 2023). The programs focused on delivering practical skills, such as gardening, cooking and social skills. A one-session VR meditation RCT was implemented for youth experiencing homelessness (Chavez et al., 2020). Overall, benefits were observed in self-efficacy, social competence, support and anxiety. These were sometimes supported by large score changes but sometimes failed to achieve significance. Thus, there appears to be limited support from these three studies, which may be due to their small sample sizes (n < 20).

## Older individuals (k = 6)

VR field trips to natural and landmark sites and exercise games offered short-term benefits for older individuals with and without disabilities. For instance, they increased observed positive and decreased negative affect (Brimelow et al., 2020; Brimelow et al., 2022; F Chaze et al., 2022; Matsangidou et al., 2023). In a longitudinal study, older individuals with disabilities showed a decrease in depression, but this finding was not robust (Afifi et al., 2022). Older women had improved physical strength and clinically significant changes in mental health following exercise games, but these were significant on a subscale of measures only (Lee et al., 2015). Notably, one study also showed increased signs of agitation in the participants after using VR (Brimelow et al., 2022). Therefore, the use of VR as long-term support for the mental wellbeing of older individuals requires further evaluation.

## Underrepresented genders and sexualities (k = 8)

*Female sexual violence survivors (k = 1).* An eight-session VR reflective writing and mindfulness meditation program was implemented for young female sexual violence survivors (Lee and Cha, 2021). Participants in the experimental group improved in perceived social support, reduced impacts of sexual violence and suicidal ideation.

*Two-spirit, lesbian, gay, transgender, queer, intersex, asexual and other (k = 7).* Studies reviewed (Acena and Freeman, 2021; Freeman and Acena, 2022; G Freeman et al., 2022; Li et al., 2023; McKenna et al., 2022; Paré et al., 2019; Reyes and Fisher, 2022) consistently reported that queer, transgender and gender-expansive youth and adults navigating social spaces experienced identity exploration and affirmation. Social VR worlds provided supportive and safer spaces through relationships with accepting individuals. VR scenarios were also used to support safer sexual practices among young men-who-have-sex-with-men by reducing shame, but the impacts were minimal (Christensen et al., 2013; Wang et al., 2021). Overall,

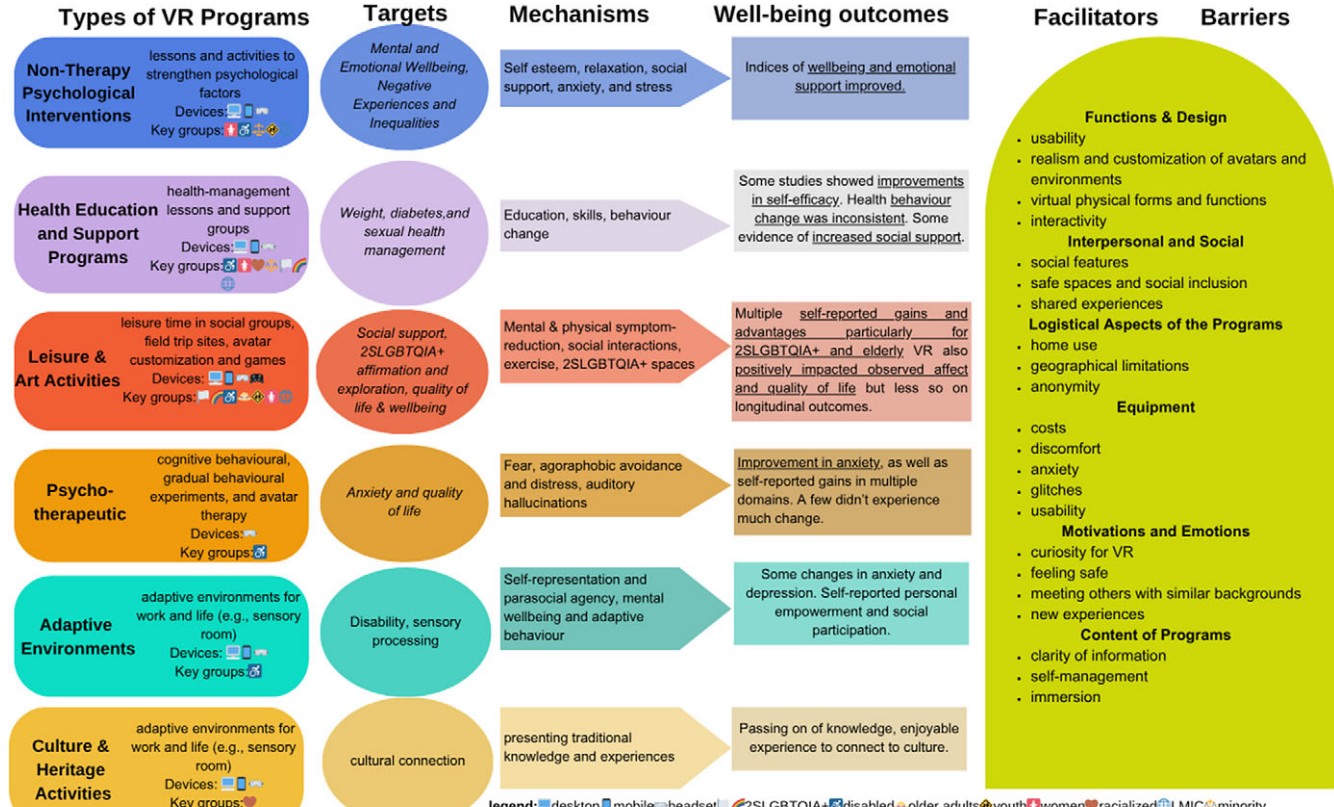

**Figure 2.** Logic model.

the current evidence for this group is primarily qualitative, which has provided a rich and descriptive overview of experiences in VR.

### Individuals with disabilities (k = 18)

Programs for persons with disabilities typically involved multi-session interventions focusing on self-efficacy, mental health, quality of life, self-management and health behaviors (Johnson et al., 2014; Singh et al., 2017; Davis and Chansiri, 2019; Kwon et al., 2022). A few studies observed the use of virtual worlds by persons with disabilities (Bloustien and Wood, 2016; Davis and Chansiri, 2019), and these were enabling for social and practical reasons, such as chatting with others and finding work. One study used VR for life skills training (grocery shopping, cooking, kitchen cleaning) for intellectual disability, equivalent to the control group (Cheung et al., 2022). In general, VR for individuals with disabilities had some benefits to their targets in these k = 6; however, some differences were notable among subgroups in other studies. To highlight the intersecting forms of marginalization experienced by participants, we have stratified our analyses following the SIITHIA best practices.

*Women with physical disabilities (k = 3).* Women with disabilities who used Second Life experienced improvements in depression, self-esteem and health behaviors (Nosek et al., 2016; Nosek et al., 2018; Robinson-Whelen et al., 2020). However, self-efficacy improved in only one of the two studies by Nosek et al. (2018). The studies all involved women with disabilities through consultations, but one study also held focus groups (Nosek et al., 2016), and another had facilitators with lived experience (Robinson-Whelen et al., 2020).

*Racialized persons with physical disabilities, including women (k = 3).* Interventions of eight to ten sessions with racialized women did not consistently improve physical, mental and health behavior indices, even with similar features to the other reviewed programs (Rosal et al., 2014; Mitchell and Laird, 2022; Mitchell et al., 2023). In one study, though, diabetes-related distress, self-care behaviors, dietary environmental barriers and other measures improved slightly with a mixed-gender sample, but this was lost at six months (Ruggiero et al., 2014). Thus, support for VR health behavior and efficacy interventions with racialized women and individuals with physical disabilities is currently limited. Two studies consulted their target populations, and two also culturally adapted their interventions.

*Autism Spectrum (k = 2).* A VR sensory room for the wellbeing of adults with disabilities or autism spectrum disorder improved symptoms of anxiety to nearly subclinical levels and improved depressive symptoms (Mills et al., 2023). Changes in social participation following the VR intervention were also noted (Mills et al., 2023), including another study investigating self-guided sessions in social VR (Stendal and Balandin, 2015). VR social worlds facilitated and empowered communication with peers and showed the potential of VR (Stendal and Balandin, 2015; Mills et al., 2023).

*Severe mental illness (k = 4).* A VR anti-stress program for persons with mood disorders helped with mindset, coping and relaxation, depression, anxiety and stress levels (Shah et al., 2015). Various VR therapies aimed at improving anxiety and increasing the quality of life of people diagnosed with schizophrenia were effective for some and showed moderate clinical significance sustained at follow-up (Altunkaya et al., 2022; Beaudoin et al., 2023; Bond et al., 2023; D. Freeman et al., 2022; D Freeman et al., 2022).

### Immigrants (k = 1)

A dissertation study (Abal, 2012) investigated how Second Life could improve anxiety for immigrants and found that the control

group reported a higher mean in anxiety than did the experimental group during learning experiences.

### Putting findings together: Logic model

Figure 2 shows the final iteration of the logic model and provides insights from the studies. For groups who may experience stigma or isolation, such as 2SLGBTQIA+, elderly and disabled persons, social VR in group settings was supportive. The adaptability of VR was beneficial for disabled persons. Non-therapy psychological interventions were applied to multiple diverse populations, in comparison to psychotherapeutic ones that had a narrower application. The study authors identified various mechanisms behind their interventions. Many programs involved education, skills training and psychological support to improve efficacy, mental wellbeing and self-management. Leisure, culture and art programs harnessed natural activities like socialization, exercise and traditions to boost social support, quality of life and other psychological factors.

Some studies provided qualitative data on the usability and design (Figure 2). No single factor was ubiquitous in facilitating or impeding VR's use. Using VR at home presented both barriers and facilitators, such as interruptions from others, greater accessibility and convenience (Johnson et al., 2014; Nosek et al., 2016). The possibility to meet people anywhere and anonymity were also facilitators (Nosek et al., 2016; Wang et al., 2021; Chaze et al., 2022; Freeman and Acena, 2022; Li et al., 2023). However, unclear information presented a barrier, while self-guided and immersive modalities were enjoyed (Acena and Freeman, 2021; Bond et al., 2023; Chaze et al., 2022; Dawson et al., 2011; Freeman and Acena, 2022; G Freeman et al., 2022). Different emotions motivated VR use, such as feeling safe, being curious about the technology, wanting to meet others with similar backgrounds and being interested in experiencing new things (Johnson et al., 2014; Stendal and Balandin, 2015; Nosek et al., 2016; Lee and Cha, 2021; McKenna et al., 2022; Mitchell and Laird, 2022; Li et al., 2023; Mitchell et al., 2023). Social VR and group interventions facilitated the creation of safer spaces and fostered a sense of shared experience. However, not every space was safe, and sometimes norms or structures that govern interactions were disregarded by some users

(Nosek et al., 2016; Freeman and Acena, 2022; Li et al., 2023). While some participants appreciated more realistic avatars, gestures and environments, others felt the deviations from reality provided emotional safety and novel experiences (Abal, 2012; Johnson et al., 2014; Bloustien and Wood, 2016; Nosek et al., 2016; Acena and Freeman, 2021; McKenna et al., 2022; Reyes and Fisher, 2022). Gender-bending and pushing the limitations of disability were also possible in VR (Acena and Freeman, 2021; Bloustien and Wood, 2016; Davis and Chansiri, 2019; G Freeman et al., 2022; McKenna et al., 2022; Reyes and Fisher, 2022; Stendal and Balandin, 2015). Some studies, particularly those providing psychotherapy, did not report any barriers. This represents a knowledge gap due to a power imbalance between providers, research teams and clients and can be resolved with mixed-method designs.

### Appraisal of equity and intersectionality

The programs' equity and intersectionality were impacted by many factors, including the composition of the research team, study planning, knowledge mobilization, eligibility criteria and ownership (see Figure 3 and Supplementary Material). Only seven study teams disclosed their positionality. In approximately half of the studies, researchers engaged with the populations being researched. The methods were typically not described in detail but included co-design, consultation, participatory and qualitative research, cultural adaptation, population-led design, or combinations of these methods. The most common method was consultations with the key community, with or without other methods. About half of the studies explicitly framed their studies on general equity and rarely intersectionality. A majority collected information on two or more axes of marginalization (k = 26); however, this rarely translated to analysis (k = 4) or interpretation. The studies overall did not conclude with any discussion of intersectionality, equity or policy suggestions (k = 1). Individuals were sometimes excluded without lifetime illness histories (k = 19), speaking a dominant language (k = 13), access or ability to use technology or the internet (k = 11) or clinic referral to study (k = 11). However, in some studies, they could self-refer (k = 11), self-define gender (k = 5) or racial or ethnic background (k = 4), and did not need proof of educational, income or housing status (k = 3), which facilitated participation. Some VR

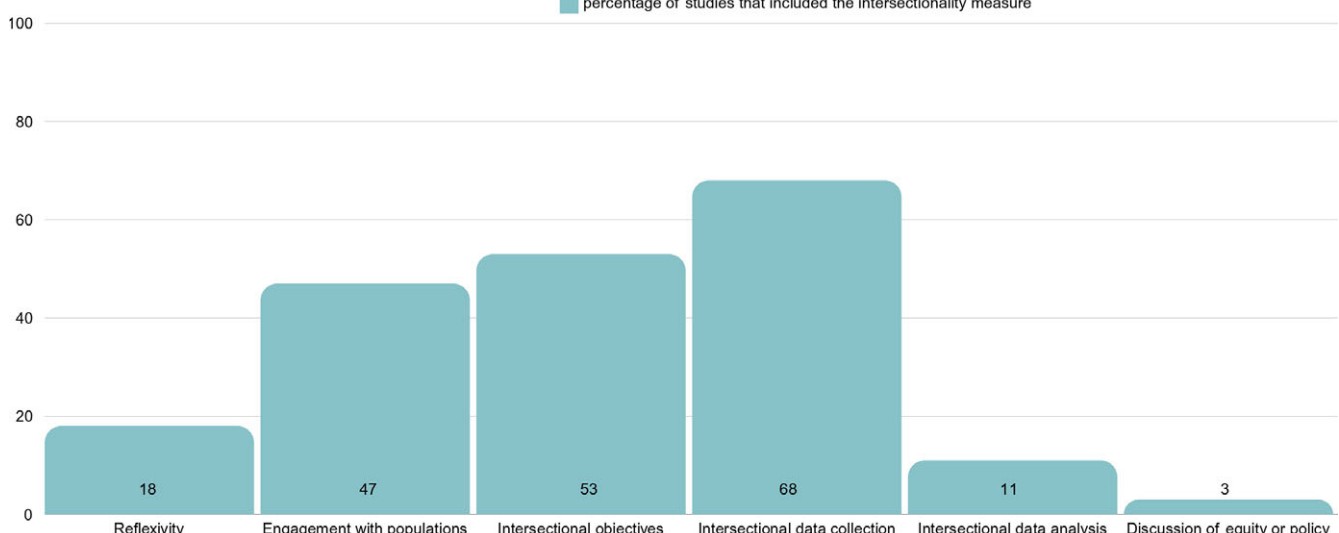

**Figure 3.** Intersectionality appraisal based on shortened SIITHIA checklist.

programs were commercially available, but few programs were entirely free (no advertisements, tiered pricing or memberships), open access, owned or continued by communities researched (see Supplementary Material). Three studies were conducted in upper-middle-income countries with youth and adults. The quality of the studies and results was heterogeneous in the LMIC context, while high-income countries benefited from the most studies.

## Discussion

In this scoping review, we comprehensively synthesized 38 VR programs from the literature. Many of the studies included populations that were intersectionally marginalized. The services provided were diverse, such as social support and affirmation, support for mental and emotional wellbeing, education, disability, physical health and cultural connections. VR opened worlds of natural, customized support that met people where they were at. VR's ability to demonstrate and present knowledge through visuals boosted participants' skills and efficacy. While enjoyment, engagement, and some benefits were evident, further research is needed to strengthen the evidence for effectiveness and efficacy. Although there was a variety of programs, some commonalities, like social and adaptive features, benefited multiple groups. Considering equity and intersectionality in the design, implementation and continuation of the projects could have widespread benefits. However, this remains a significant gap in current practices.

### *Leveraging naturalistic environments*

Our findings showed that even programs that were not based on psychotherapy techniques like cognitive-behavioral therapy could still leverage naturalistic benefits and achieve successful engagement. For example, programs with cultural elements passed along important heritage and encouraged reconnection with culture. In VR social spaces, users of all ages sought and provided emotional and social benefits, including stigmatized groups like 2SLGBTQIA +, who created safer spaces with their own rules for respite from marginalization. The impact of leisure on subjective wellbeing is robust across countries and age populations (Kuykendall et al., 2015). Indigenous *culture as health* is a framework which states that cultural practices are not just complementary but are the basis of health, further supporting community and leisure activities linked to wellbeing (Yamane and Helm, 2022). Peer-to-peer programs often lead to improved recruitment and retention (Sokol and Fisher, 2016). Studies without structured programming can also offer participants benefits, possibly due to the natural advantages of social support (Acena and Freeman, 2021; Li et al., 2023; van Brakel et al., 2023; Wang et al., 2023).

### *Mental and physical programming challenges and opportunities*

VR enhances customization opportunities, which are perfected through co-design, testing and iterations. For instance, programs designed to build skills and knowledge for physical health had mixed results, particularly for women and racialized individuals. These studies employed various methods of community involvement, such as consultations and cultural adaptation. Some inconsistency in findings may be explained by a multitude of factors, including the small number of studies identified. However, improving the integration of intersectionality and collaboration could help

address these challenges (Davidson et al., 2013). A starting point for many digital mental health (DMH) interventions is the standard of care for mainstream populations, which is then adjusted to encourage use by marginalized populations (Hwang, 2009). However, this carries the mainstream's norms and assumptions about the efficacy of the approach for marginalized groups (e.g., individual preferences about VR) (Schueller et al., 2019). Several skill-building programs we reviewed were built bottom-up, but very few provided in-depth qualitative inquiry about preferences and no information about preference for VR versus other programs. Strategies such as addressing structural barriers, endorsement of marginalized groups through race-matched facilitators, collaborative working and gender considerations are pertinent (Davidson et al., 2013). Following this, further testing and iterations should ascertain that the modifications are appropriate and retain their meaning. Lastly, improving the quality of the literature with larger, controlled studies is essential (Mathews et al., 2019).

The VR programs targeted primarily internal factors, like anxiety, positive and negative affect, depression, gender exploration and affirmation, health-related distress, quality of life, social support, and suicidal ideation. However, as several reviews note, recognizing the condition (e.g., health symptoms) and not the cause (e.g., marginalization and social determinants of health) sometimes strays from the principles of intersectionality and equity (Buchanan and Wiklund, 2021; Alegría and Cheng, 2023). To truly have a positive impact on wellbeing, VR programs must contribute to the transformation of the systems of oppression that reinforce each other (Baum et al., 2009; Meskó et al., 2017). VR can be a change-maker. For example, its accessibility to the public rebalances power dynamics. Persons with lived experience become creators. This replaces mainstream norms with a community-led response to wellbeing. There are also opportunities for cultural plurality and increased access to care through culturally tailored or translated triage and psychoeducation programs (Pendse et al., 2022). Indeed, as Carey and Crammond (2015) write, "*it is evident that the power of an intervention comes not from where it is targeted, but rather how it works to create change within the system*" (Carey and Crammond, 2015).

### *Equity and intersectionality pitfalls and solutions*

We identified some challenges related to equity and intersectionality and will propose solutions in this section. The results from the modified SIITHIA intersectionality checklist (Blair et al., 2022) indicated an effort to involve communities with room for improvement. Firstly, the studies fell short of conducting intersectionality analyses, which undermines the representation of diverse experiences and knowledge mobilization in future research. Secondly, European binary/cisgender gender populations were overrepresented, and certain racialized groups and genders were underrepresented. Participants were often excluded from studies for not speaking a dominant language, lack of internet or technological access, or having specific illness histories or diagnoses. This may reflect a systemic inaccessibility to research participation (McCall et al., 2021). Inclusion should be improved by providing all necessary materials and training, balancing convenience sampling from clinics with other recruitment methods and offering appropriate remuneration for the resources and time to participate (McCall et al., 2021). For the mental and social wellbeing of the populations being studied, it is essential for privilege to be rebalanced in recruitment practices, led by an understanding of financial or other privileges and destigmatization (McCall et al., 2021). While

technology is mainstream, access varies within and between marginalized populations and countries. Thus, widening access to the internet and digital technologies in close consultation with affected communities and incorporating intersectionality theory remains a recommendation (Labrique et al., 2018; Husain et al., 2022). VR is not always the appropriate technology for every setting or desired by every community. But further research is needed to examine how VR perpetuates and amplifies inequities, such as privileging higher-income, English-speaking communities (Jardine et al., 2024).

A lack of leadership support for marginalized groups is a key challenge. Cultural norms are often imposed in healthcare, and thus, some knowledge is discounted (Armaou et al., 2020; Pendse et al., 2022; Radu et al., 2023). Nevertheless, VR can recognize and address cultural forms of distress and incorporate the positive psychology of each local context (Craven et al., 2016; Mendenhall and Kim, 2021; Pendse et al., 2022; Silverman et al., 2023). Thus, a genuine community desire and putting aside assumptions or imposed norms must precede the commencement of a project. Community consultations, relationship-building and other strategies are part of an in-depth knowledge-gathering process. Furthermore, other researchers can leverage their privilege to improve access to plural forms of care (Pendse et al., 2022) by assuring open access to their programs for modification or use. Indeed, ownership and continuity of programs represent a significant challenge. Most of the programs reviewed were developed and owned by researchers and never made available to the communities researched (certain exceptions in the Supplementary Material). As the selected articles spanned over a decade, the relative lack of permanent digital wellbeing infrastructure demonstrates the research-to-practice gap identified elsewhere (Morris et al., 2011). This implementation gap demonstrates a weakness in the feasibility and cost-effectiveness of the programs developed thus far; however, future work should address these challenges. Ownership by marginalized groups seemed to leverage naturalistic benefits, such as connecting with others over lived experiences and social support.

Additionally, marginalized individuals found cost-effective solutions around issues of ownership and project continuity in Second Life. Researchers have made suggestions similar to these VR users' projects; Mohr et al. (2018) say to *create, trial and sustain* digital tools to increase their feasibility and utility (Mohr et al., 2018; Schueller, 2021). Thus, we encourage stakeholders to incorporate translation strategies, open access and data sovereignty within their VR and digital health programs (Armaou et al., 2020; Pendse et al., 2022; Radu et al., 2023). Low-cost solutions or project end dates could be among the solutions researchers use to overcome funding barriers limiting open-access initiatives. A future direction may be to perform an environmental scan to identify challenges like VR program ownership and other equity issues.

### Recommendations

VR interventions are enjoyable and relevant to marginalized populations, but the literature on their impact and effectiveness was quite heterogeneous. Our logic model (Figure 2) summarizes these findings and their generalizability across populations. We identified limited studies meeting our criteria for wellbeing and not just targeting symptoms. Targeting symptoms or conditions is more appealing to some populations (Boucher and Raiker, 2024), sometimes reducing attrition. However, the context and healing paradigms of the populations are important (Borghouts et al., 2021). The literature speaks to how co-development and leadership from

marginalized populations benefit the implementation of wellbeing programs (Cyril et al., 2015). Thus, we recommend the following:

- To increase the strength of evidence, stakeholders might focus on replication studies and adapting existing VR environments. This will allow time for centering the populations in the research process. It also allows time for thoughtful planning and assessment of the population's context, relevant issues and healing paradigms.
- Stakeholders should employ intersectionality theory and equity-based approaches to parse out the variability in preferences about VR (Cyril et al., 2015).
- To reduce barriers to VR, opportunities for privacy when using the programs and setting group norms are universal adaptations.
- Also, it seems participants may prefer to have a sense of immediate success of benefit (Borghouts et al., 2021). A brief psychoeducation and goal-setting module could help train participants to use the technology and learn what to expect from the intervention (Boucher and Raiker, 2024; Mukhtar et al., 2025).
- In general, interventions using goal-setting techniques can improve outcomes (Stewart et al., 2022).
- Reflecting on biases and positionality is important as there are significant digital divides for persons of the global majority (global south) and within high and LMIC contexts (Tsatsou, 2011).
- Incorporating population leadership or participatory research alongside mixed methods will help researchers to assess systemic issues affecting their project (Stiles-Shields et al., 2022). For example, this should involve prioritizing underserved populations, establishing whether the population has basic access needs, correcting epistemic exclusion by prioritizing Indigenous healing paradigms, and other strategies to transform health systems with digital technology (Ramos et al., 2024).

### Strengths and limitations

We addressed the knowledge gap on VR use by marginalized groups. Moreover, we took a complex approach by examining equity and intersectionality in research design, systemic factors and program features. We have also provided recommendations, including addressing the issues of power to stakeholders. Equally, we identified a key gap: subgroup analyses were rare, so were intersectional interpretations. Our search strategy was verified by members of the research team and a university librarian. However, we excluded articles that were symptom-centered and about non-marginalized populations, which limits this review's scope. We manually quality-checked; however, our AI reviewer has a potential 5% margin of error. Additionally, positive research findings tend to be reported, while negative findings are not, meaning a potential publication bias. The included studies often had smaller sample sizes, reflecting the possibility that this literature is in development for marginalized groups and groups unrepresented in this review, such as lower-income countries. Though several included studies were RCTs, effect sizes and practical significance were underreported. Due to the heterogeneity of measures, we interpreted findings primarily through narrative synthesis. Researchers of diverse backgrounds and expertise compose our team; however, we may have biases due to our positionalities. Equally, the premise of our question may reflect privilege and differences in experiences, which limit its utility to some populations. The studies focused on communities using VR but not those ambivalent, against its use for their

wellbeing or who lacked access to VR, except in a few studies, which is a limitation of our findings.

## Conclusion

VR programs developed for and by marginalized populations have many applications and benefits. We provided a logic model, examining factors for various kinds of VR interventions, and provided their target populations and technologies used. Marginalized researchers and community members can also find a list of publicly available programs reviewed in our supplementary content. Future directions are to conduct subgroup analyses, ensure the continuity and ownership of programs by the populations studied and address underserved populations like racialized, Indigenous and gender expansive communities. In future studies, researchers should design with the intention of transforming health systems to address equity for marginalized populations, especially through collaborations and population-led designs.

**Open peer review.** To view the open peer review materials for this article, please visit http://doi.org/10.1017/gmh.2025.10084.

**Supplementary material.** The supplementary material for this article can be found at http://doi.org/10.1017/gmh.2025.10084.

**Data availability statement.** The data that support the findings of this study are openly available on the Open Science Framework. A pre-print of this article and its contents was made available at https://doi.org/10.31234/osf.io/3u7ea.

**Acknowledgements.** Many thanks to Jill Boruff from the McGill University library for advising us on the search strategy.

**Author contribution.** Conceptualization of the study and methodology was in collaboration between Q.S., Martin L., Myrna L. and G.S. The search and screening team who were involved in the investigation were A.H.M., E.C.D., L.S., C.J. and C.D. Software testing and modifications were done by Q.S. and C.D. Data were validated (Q.S., C.L. and A.H.M.) and then analyzed by Q.S. Q.S. is the main author, writing the original draft, which was reviewed and edited by all authors.

**Financial support.** This work was supported by a doctoral fellowship (https://doi.org/10.69777/316179) from FRQS by the primary author (Q.S.). Otherwise, this research received no specific grant from any funding agency, commercial or not-for-profit sectors.

**Competing interests.** The authors declare none.

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
