## [Reviewer Report]

Overall this is a really important topic and review. I was asked to comment on how well global content was included and discussed, and this is a significant limitation I note below that needs to be explicitly addressed. I believe with more specificity and attention to study design/purpose, this review could be improved and have great implications for research and practice. Specific comments below:

Impact statement:

-Consider a range of socio-demographic factors that are know to be associated with marginalization and healthy inequities -currently “How can sex, gender, race, and ethnicity be better included in the design and research on this topic?” yet there are other important social determinants of health, including sexual orientation, dis/ability, immigration status, class and income; and also global diversity. From an intersectional lens rooted in the Combahee River Collective, class, race, gender and sexual orientation are inextricable from intersectional analyses.

-this can be reworded; are you focusing on positive psychology, salutogenic factors (e.g. this is not clear: (Could further community engagement help identify non-symptom-based, positive, or other relevant targets and mechanisms?)

Abstract

-need to include more detail on how marginalization showed up in the findings: what kinds of populations or health issues were included? What global regions (including low and middle income contexts-LMIC)? How much racial/gender/SES/sexual orientation and regional/LMIC inclusion in your 38 articles?

-most studies reported ‘positive’ outcomes: please be more clear what this means

-also consider noting the digital divide and digital literacy as key factors relevant to LMIC inclusion in VR

Main paper-introduction

-this could be strengthened by including clear examples of who you are really talking about when you say marginalized people. What marginalized people have been using VR and what ones are still not using it? There is a generic marginalization that is hard for the reader to understand

-there is also no geographic specificity-is this mainly in high income contexts vs. LMICs?

-line 115 gives some good examples but no geographic locations are listed, please add more specificity and provide the reader a better understanding of study designs used to make the claims

-was there a librarian involved in the search strategy

-screening: how did you operationalize this marginalization criteria (populations that were marginalized, defined as having limited access to health promotion and more exposure to health risks) in your search?

-your positionality experiences could be in an appendix and this would allow more space for more detail in the preceding section about the background literature in this area per above, as well as more detail about how you operationalized your key variables.

-in particular, how you categorized marginalization is really important to describe for the reader and how did you make that decision/definition

-since only 11% of literature was from low and middle income contexts, I recommend adding another section/subheading along with the other populations you describe entitled ‘low and middle income contexts’ (LMIC) and detail where and how this was used, and any results. This is really important as there is a serious digital divide in VR and you have the opportunity to showcase what has been done in LMIC and discuss if it was different from high income countries in any way and how it was culturally/contextually adapted. Without this there can remain a bias in your review towards showcasing high income examples

-the logic model can also disentangle the findings by high income vs. LMIC as there are different barriers and facilitators to access and digital literacy

-appraisal of equity and intersectionality: the authors state that ‘only 7 teams disclosed their positionality’ but this is not a required tenet to show equity -there is not a standard practice and depending on where people publish, it may not be accepted. So this could perhaps be a recommendation (if you have literature that it matters if people publish their positionality); another consideration is safety for co-authors to disclose publicly a stigmatized identity in contexts of criminalization and stigma and discrimination-the assumption that everyone can and should publicly disclose their identities is not fully agreed upon in the equity field so should not simply be stated without any critical reflection on the privilege and safety (particularly thinking of where lgbtq folks, sex workers, people who use drugs, people with HIV are criminalized)

-this section needs a bit more nuance: participants being ‘barred’ from participation is framed in a negative way rather than as inclusion/exclusion criteria within a scope of study

-discussion: when you can more fulsomely describe how you define marginalization this can be updated to reflect that, and when you disaggregate the LMIC vs. high income studies this can also be integrated in the discussion

-the first paragraph of the discussion seems to present a more strong view on the efficacy of VR than was presented in the results, so I suggest specificity in what was actually efficacious vs. enjoyable (both important, but different with different implications)

-the discussion should not be presenting new information/results, so some of the mental and physical programming challenges and opportunities needs to be integrated back into the results, and then briefly referred to here but no new results in the discussion

-there is a discussion in the tables of who included intersectionality measures: there is a field of research on this that needs to be more fulsomely referenced in your methods section: how did you assess if intersectionality was measured, what references did you use etc. What may be more accurate to report is, if the studies were designed to address intersectionality and thus measure it, or if you are looking at studies for what you think they should have done (rather than assessing what their objective/aims were). It is possible that they addressed equity but did not apply an intersectional lens, so how you frame/critique studies should also take into consideration the very purpose and scope of the study otherwise you are doing an extra critique that is beyond what the study ever meant to do

-equity pitfalls also presents new findings: please separate findings from the discussion so discussion is situating the findings you already discussed within the larger literature

-also think about the limitations facing researchers; maybe they do not have funds forever to make the VR program open access and face other barriers that are structural

-I think both tempering your language and assumptions about what studies should have done, and being more specific about what they did and found, will strengthen the paper. Considering the strengths of the studies within their scope, and more generously discussing ways forward, will help your paper feel less judgmental toward researchers and more productive/generative

-the conclusion is too long and should be a high level summary of what was found and what we should do next (some of the conclusion can be integrated into the discussion)

-overall more specificity and acknowledgment of limitations of a single study should be considered; even the last sentence ‘Equally, researchers should design their studies around transforming health systems to address equity for marginalized populations’ does not provide any examples or a roadmap of how a VR study can transform a health system to address equity, being more clear and with grounded examples of what that could look like would help the reader

-as this is a global mental health journal, exploring and discussing implications of your review in diverse global contexts (including LMIC) is key for this specific journal focus and for the field at large.

---

## [Reviewer Report]

The authors of “Virtual Reality Offerings for and by Marginalized Populations: A Scoping Review on Equity and Intersectionality” set out to identify thematic practices of well-being instituted by and for marginalized populations via virtual reality (VR). The article is ambitious but currently falls short of developing novel and productive insights. Indeed, the work may have been scoped too much, too stringently, as valuable scholarship from IEEE and ACM databases is not really present. These are the two most popular databases for VR design and development scholarship. Not including these scholarly databases was a major misstep in the work. The authors say Google Scholar was used, so these databases might be present, but only one citation is attributed to each. The authors would find many efforts of VR designers working to engage and help marginalized populations in those archives. The authors are encouraged to go back and include these archives in their approach.

Beyond this, there are a few sections in the article that lack clarity. It begins with the title, which mentions this is a scoping review of intersectionality and equity. However, the article focuses on “wellbeing” and “well-being” (used interchangeably and present in both forms throughout the manuscript—further, the term is not defined). For example, in the abstract, "we performed a scoping review of VR’s use for wellbeing." However, well-being is not mentioned in the title. A standard or citable definition for well-being would help anchor and frame the work. Later, the work discusses equity in the objectives and then on page 11.

The introduction could be better focused on establishing the problem space and active research streams. Here, a definition of well-being could be presented, and a deeper discussion of how well-being is intersectional would be productive. Further, the authors might clarify whether the focus is on the “Equity and Intersectionality” of marginalized communities' uses of VR for well-being. This kind of clear statement may be the bridge that is missing.

As a VR researcher, I am concerned that some of the Second Life scholarship is addressing the non-VR version of the platform. The non-VR version of Second Life was often called a “virtual reality” but was most frequently not experienced in a headset, like contemporary VR. The authors should either define a comprehensive definition of VR that encompasses non-headset examples or check that these examples were indeed in VR based on our contemporary understanding.

The discussion of the inequities related to VR usage needs more attention. It is glossed over in the Objective section. Honestly, the inequities spread across all versions of XR along the virtuality continuum. The authors are encouraged to explore these inequities through the lens of spatial justice.

Readers would likely appreciate a table with the pre-existing literature filters. The authors' tables and figures are very helpful!

Other sections could be cleaned up. Removing anti-prejudice articles is odd and not well-justified for an article focusing on equity.

In the discussion section, the authors discuss leveraging the natural environment but what follows is more about socio-cultural environments. Interestingly, there is research in the ACM and IEEE libraries about how VR green spaces encourage well-being. The authors should engage with this literature to improve this discussion. Further, there are a few claims in the discussion that have a single citation given. This approach seems incongruous as the evidence for thematic claims was well-cited in the results section.

In the end, the authors should revise primarily for clarity but should run their selection process through the IEEE and ACM proceedings before doing so. This would likely improve the novelty of their insights, and progress could be made more actively in this critical domain.

---

## [Reviewer Report]

Introduction:

Including a more detailed definition of “VR” would be a benefit for readers who are not familiar with simulation technologies. Perhaps, even more helpful for readers is a description of “marginalization” and “marginalized groups.” Additionally, I think defining “wellbeing” and “adaptive approaches” would help readers better understand and appreciate the article.

The reviewer appreciates the inclusion of intersectionality into the introduction, further elaboration would be useful to those who are not familiar with the term, and the authors should explain what they mean by “intersectional marginalization.”

Language and choice of words are important when talking about people who have been traditionally marginalized and researchers who do not identify with these communities need to be particularly mindful of tone. For example, “marginalized persons are currently under-served by VR and other digital health for healing and support when desired”: what do the authors mean that people from underserved groups are underserved by technology, which is created by people? And who is judging desire and whose desire? Rather, for example, would it be more accurate to say that “members of marginalized groups are finding few VR and other digital tools for purposes of healing and support when they are seeking them?”

“To date, VR is not being fully applied to the benefit of marginalized groups, who access a dearth of healthcare resources and are exposed to greater wellbeing risks” is unclear to this reviewer. Do authors mean that marginalized groups do not access healthcare resources? Or that there is a relative lack of healthcare resources for those who identify with marginalized groups?

In the second paragraph, specific examples would help the reader understand the researchers. What does “culturally relevant modalities” mean? What does “plural forms of care” refer to? And what impacts might they have?

There are key concepts in the second paragraph which are well known to those familiar with simulation-based health professions education, but warrant defining to facilitate a wider audience (i.e., embodiment and immersion).

In the third paragraph, the authors name challenges that face VR: it would be more accurate to state that there are challenges for the design, implementation, access, or reach of VR. VR is a product, and the challenges are faced by its creators (and those who implement or use it to teach), not the modality itself. These challenges should also be defined for the benefit of the readers: what do they mean by “systemic inaccessibility” and “monolithic programming”? While the authors offer an example of Indigenous participants’ experiences, it is not clear if the example speaks to one of the named challenges or all of them. It is important to remember that Indigenous Peoples constitute members from many nations and their views will likely be heterogeneous. Again, terms such as “conventional care” should be defined: are the authors referring to healthcare? Mental healthcare? Western healthcare? This is germane for this publication forum, which specifies “global mental health” and should try to avoid asserting a white North American/European dominant narrative. The authors also suggest that demographic information and diversity constitute diversity, equity, and inclusion: again, here, this reviewer feels that the current text would benefit from elaboration.

Objectives:

This section would benefit from re-writing. What does “reproduction of inequities in some forms of VR” mean? Does this refer to the unintended result that VR can replicate or even possibly magnify the health inequities which currently exist? The authors should discriminate between “wellbeing” and “strict clinical goals”: how are these two different and would there be a situation where these two goals overlap? The aims should be clarified: what is a target and an outcome? How do the authors define these constructs? And from whose point of view?

Methods:

The reviewer benefited from referring to the study protocol registered on OSF Registries and suggest that including the section “The review will include…” and the exclusion criteria in their paper will add clarity for readers.

This reviewer appreciated seeing their logic model, and suggest that they reference the logic model earlier, as this would have clarified the objectives.

This reviewer appreciated the statements of positionality from the researchers. While the researchers have clearly experienced various forms of marginalization, they acknowledge their privilege in terms of their educational/employment background and that they live in a high income, Western country.

Given that this scoping review is examining VR created FOR marginalized groups and BY marginalized people, using an equity lens, the research team could be strengthened by including: a member of members of marginalized groups who are actual users of a VR program; and a member of members of marginalized groups who is an actual developer of a VR program, designed for their own group or another marginalized group. Inclusion of members with lived experience may strengthen the design of the study and the validity of their findings.

The reviewer appreciated the authors’ efforts to ensure that their research activities were inclusive and equitable, which included referring to and using the SIITHIA intersectionality checklist.

Results:

A more detailed discussion earlier in the Objectives or Methods section around “wellbeing” and “strict clinical goals” would be helpful, as in the Results sections, the authors include clinical outcomes such as “decrease in depression” and “improved physical strength” which could be considered clinical goals from the users’ perspective.

Discussion:

A rich discussion section.

I would have liked to see more elaboration around co-design and co-creation with members of lived experience for whom a VR is being created by someone who does not share that experience, which is often the case for simulation-based interventions. True co-creation and testing would assist in identifying areas that could be iteratively improved for end-users, such as unclear information, intuitiveness of the program, realism, etc.

The authors indicate that VR can be a changemaker, though its accessibility and potential to redistribute of power towards those creators with lived experience. However, it should be acknowledged that VR can amplify power differences, through the privilege of technology. For example, those who experience housing insecurity and poverty are less likely to have access to VR technology and to have the knowledge and skills to use the technology as a creator. Efforts must include making technology and VR more accessible to people with lived experience.

The authors describe representation within the programs they reviewed and indicate that certain groups were overrepresented, while others were underrepresented. Additionally, some regions are clearly overrepresented, such as the United States and Canada, and the programs are predominantly English. These trends reflect the research and development of VR technology, and the use of VR may amplify these forms of overrepresentation. And some groups may be overrepresented in certain regions. For example, cisgender/binary gender populations may be overrepresented in the US, Canada, UK, and Australia but may be completely absent in other parts of the world, which could be of interest for a global publication.

This study looks at what is publicly available through systematic search of the literature and the authors indicate that most of the programs they reviewed were developed and owned by researchers. They indicate that ownership and continuity of programs are challenges, which also include that some programs are proprietary and require users to pay to use them, which forms a barrier to access. This is another mechanism by which VR may amplify differences of privilege.

Furthermore, the authors note barriers encountered by users to access such programs, but these deserve more discussion in the context of a global review, such as: users’ ability to access to technology; reliable internet access; users’ ability to access the internet; users’ ability to use VR technology. These are relevant factors, as computer access and reliable internet services are not universally available. Even within a high-income country as Canada, northern areas of the country do not have reliable internet. And these factors are issues of equity and privilege which may not be strongly evidenced in the programs reviewed by the authors; they are likely barriers encountered by the non-users, which may account for their non-participation in the programs reviewed and lack of representation amongst users. A limitation of this study is the lack of the voices and perspectives of the non-users.

---

## [Editor Report]

Thank you for submitting your manuscript for review. Although the reviewers acknowledge the relevance of the subject, they have identified notable flaws in the background, methodology, findings, and their interpretation. The reviewers have provided useful recommendations that could improve the manuscript. We invite you to carefully consider and address the reviewers’ comments and recommendations and resubmit a revised manuscript.

---

## [Reviewer Report]

INTRODUCTION

The Introduction opens with “[v]irtual reality (VR) could improve wellbeing, the composite of across multiple dimensions, of marginalized groups, if done so equitably.” At least four concepts or constructs should be defined: VR; wellbeing; marginalized groups or marginalization; and equity. The definition of VR should be revised: VR does not create a 3-D world but rather IS a simulated, 3-D world which is created by using computer technology, and with which users interact and experience through equipment that provides sensory stimuli to users. Furthermore, in the Inclusion criteria section, the authors indicate that studies needed to have “met the definition of VR from Sherman and Craig”; their definition could be included in the Introduction.

“Wellbeing” is not clearly defined in the Introduction: is wellbeing synonymous to relaxation, encouragement to adopt health behaviours etc.? The authors clarify this term in the Inclusion criteria section; however, I think it would help the readers to have this earlier in the Introduction section.

“Marginalization” is also not clearly defined by the authors in the Introduction. Further clarity around this concept would be of benefit to readers, especially since “marginalized” has been contested as being stigmatizing. The AMA/AAMC suggests, for instance, that “groups that have been economically/socially marginalized; groups that have been historically marginalized or made vulnerable; historically marginalized” are preferred terms as they are more equity-focused (American Medical Association and the Association of American Medical Colleges (AAMC) Center for Health Justice “Advancing Health Equity: A Guide to Language, Narrative and Concepts,” available at https://www.ama-assn.org/system/files/ama-aamc-equity-guide.pdf).

I think the authors should further clarify the concept of intersectionality, especially since that they suggest that it is “a key consideration for VR creators”: the result of experiencing multiple forms of oppression, concurrently, is a unique, subjective experience for an individual and the collective result of experiencing multiple oppressions is not merely additive. I also feel the following sentence “[i]ndividuals become caught in the margins as multiple systems of oppression work together,” can be interpreted that “being multiply oppressed” is a chance occurrence, whereas systemic racism is a result of deliberate policy and political choices made by dominant groups to deliberately oppress other groups. Groups are marginalized not only through social processes, but also economic and political processes.

Please clarify: “This dearth of resources is robustly associated with experiences of racism, heterosexism and other marginalizations”—experiences of a certain group of people? Certain groups of people? If there is an emphasis on intersectionality, identifying that a particular group of people are experiencing specific, multiple, concurrent oppressions would be helpful.

I wonder if “Indigenous participants…indicated that digital health did not consider their needs…” speaks more to that Indigenous health is simply ignored or absent in digital health, as in many ways, representation of Indigenous Peoples is missing in simulation-based medical education.

OBJECTIVES

Please clarify who are “marginalized stakeholders.”

The authors have included a statement of positionality in the Methods section. I wonder if this section should be brought up earlier, as I feel that the diversity of the team and the lived experiences of team members, who include “marginalized VR users and creators” have shaped the objectives of this project. I think that including a discussion of “allyship” (The Anti-Oppression Network, available at https://theantioppressionnetwork.com/allyship/) and “advocacy” as researchers and authors would contribute to the richness of this paper.

RESULTS

Please consider re-categorizing the program for young female sexual violence survivors. Literature suggests that intimate partner violence is experienced by a large proportion of women and girls and should not be characterized as a “minority” experience.

Furthermore, the use of “minorities” is controversial, as it may seem like as an example of “othering” language and “sexual and gender minorities” may imply cisheteronormativity as the desired dominant standard.

“Individuals with Disabilities” section offers an opportunity to refer to intersectionality and would present the reader with concrete and relevant examples of how forms of oppression and marginalization can intersect.

Putting Finding Together

“Different emotions motivated VR use, such as feeling safe, being curious about the technology, wanting to meeting others with similar backgrounds, and being interested in experiencing new things.”

“Social VR and group interventions facilitated the creation of safe spaces and a sense of shared experiences. However, not every space was safe and sometimes norms or structures around

interactions were not followed by some users.” There is discussion about what constitutes a safe space, and who determines “safety” of a space: for example, https://www.tandfonline.com/doi/full/10.1080/14675986.2019.1540102#abstract

DISCUSSION

Leveraging naturalistic environments:

Please define “un-manualized” programs.

Mental and physical programming challenges and opportunities:

Please spell out “DMH interventions.”

“A starting point for many DMH interventions is the standard of care for mainstream populations, which is then adjusted to encourage use by marginalized populations (Hwang 2009). However, this carries assumptions about the efficacy of the approach for the marginalized groups; for example, individual preferences about the use of virtual reality (Schueller et al. 2019).” This also reinforces the idea that the dominant group is the norm.

Suggest editing to “The VR programs targeted primarily internal factors, such as wellbeing, anxiety, affect, depression, gender exploration and affirmation, health-related distress, quality of life, and suicidal ideation. Though, as several reviews note, recognizing the condition (e.g., health symptoms) and not the cause (e.g., marginalization and social determinants of health) sometimes strays from the principles of intersectionality and equity (Alegría and Cheng 2023; Buchanan and Wiklund 2021). To truly impact wellbeing, VR programs must contribute to systems-level change and the transformation of the systems of oppression which uphold each other (Baum et al. 2009; Meskó et al. 2017).”

Equity and intersectionality pitfalls and solutions:

“Programs for wellbeing being developed by marginalized groups are a key challenge”: in what ways are they a challenge? It seems to this reviewer, that the authors describe, in this paragraph, the challenges that such programs face or encounter.

Recommendations:

Suggest revising “VR interventions are enjoyable and relevant, but the literature was quite heterogenous on the impact and efficacy” to specify the populations of interest for this study.

Strengths and limitations:

“Equally, the studies focused on communities using VR but not those ambivalent or against its use for their wellbeing (i.e., non-users), which is a limitation of our findings”: I would add that additionally, “those who were unable to use VR for this purpose (ie., due to lack of access, lack of resources) are not represented.”

CONCLUSION

The authors might consider also include suggestions that researchers should partner with members of the community of interest to engage in true co-creation, co-design and collaboration.

---

## [Reviewer Report]

The work is much improved. My main suggestion is that the authors revise the introduction which could be clarified to better establish their position and research. I think this could be done with some minor restructuring.

Instead of discussing facets of the problem before presenting the clear objectives around line 185, the authors might present a concise statement in the introductory paragraph. Lines around 246 might be elevated to clarify the entire objective of the article more succinctly. From there, the authors can then define wellbeing, the challenges to marginalized creators and audiences in VR, and marginalization generally.

I do think that in the screening criteria, clarity might be provided by defining what a “symptom-based” outcome is. I understand this after reading the paper twice, but a brief explanation here would be useful. We get some insight on this in Limitations, “However, we excluded articles that were disorder-centered and non-marginalized populations, which limits their scope.” Grabbing that line, I can see there is a typo in it. I think you mean to say “and about non-marginalized…” All the same, are symptom-based outcomes not disorder-centered? This is the clarity I seek.

The authors are also encouraged to bullet out their recommendations for clarity and readability.

---

## [Reviewer Report]

The authors have made great progress in strengthening the manuscript and attending to the extensive reviewer comments. Congratulations on this progress.

I have 3 additional recommended revisions before acceptance:

1) first, there are a lot of typos and issues with the word choices (abstract for instance, “Black or African American (26%) and European or White (53%) individuals composed many studies, but other sociodemographic characteristics were underreported”: does this mean they developed the study, or the study populations included these ethnoracial groups? the word composed is confusing here).

Also on page 11 “The possibility to meet people from any location or be in anywhere”.

-there are also a lot of terms used for sexually and gender diverse persons with no consistency or definition -to make your reading accessible please try to use consistent language or offer defintions of terms/acronyms for the global readership

I suggest a careful proof reading line by line with an Expert english speaking proof reading is needed.

2) Second, I feel the gaps regarding low income and LMIC country exclusion from the literature warrant express attention. There was a recent Ugandan study published on VR in this journal, and there is a need to really think about what it means and why it matters or not that there were no low income countries in your study. This is only briefly touched on and you mention LMIC but then do not elaborate what that means when we consider underfunded health systems, structural roots of problems, etc. As a global journal I would like to see a bit more here.

3) finally, I do not feel the explanation of systems change with VR is enough. The example lacks specificity. What about transforming systems by using VR to reduce stigma and improve cultural competence in health care, or policing etc? For me this is a bit too vague still: "To truly impact wellbeing, VR programs must contribute to the transformation of the systems of oppression which uphold each other (Baum et al. 2009; Meskó et al. 2017). VR can be a changemaker, for example, through its

accessibility to the public, which rebalances power with persons with lived experience as creators. Indeed, Carey and Crammond (2015) write that, “it is evident that the power of an intervention comes not from where it is targeted, but rather how it works to create change within the system” (Carey and Crammond 2015)." There are no examples of systems change so this can be amended

After you attend to these, I think this could be a good addition to the literature.

---

## [Editor Report]

Thank you for revising the manuscript and addressing the reviewers’ recommendations.

All reviewers are satisfied with the revisions but have requested some minor additional changes.

We kindly invite you to carefully consider and respond to the reviewers’ comments and suggestions. Please submit a revised manuscript when your revisions are complete.

---

## [Reviewer Report]

The manuscript needs editing; the authors should create a clean copy to review, as there are multiple instances where grammar needs revision, including the opening sentence of the Background section.

Background:

This reviewer suggests starting with the last sentence of the first paragraph, and then to define the concepts which are foci of this article: VR, wellbeing, and marginalized groups. The concept of marginalization is still not yet clearly defined in this current iteration of the manuscript and this reviewer’s comment that “marginalized” has become a contested term and is considered stigmatizing. This reviewer wonders why the first sentence of the second paragraph specifically identifies two distinct groups of people (“[i]ndividuals who are excluded from healthcare resources and exposed to greater wellbeing risks, or marginalized individuals”), whereas individuals who are excluded from healthcare resources and exposed to greater wellbeing risks ARE marginalized. The authors specify “marginalized” populations as “as having limited access to health promotion and more exposure to health risks” in the Inclusion Criteria, and this reviewer suggests defining “marginalization” earlier in the manuscript will help with clarity.

Readers would benefit a clearer and more accurate definition of intersectionality. The sentence “marginalization influences a person’s health status and access to health care, which are inextricably linked, as intersectionality theory describes” needs unpacking, as it is not actually obvious to readers how marginalization, health status and access are inextricably linked and how this interaction is an intersectional one.

The authors’ choice of definition of equity requires a reference. This reviewer suggests that they consider the classic definition of equity as put forward by the WHO: “Equity is the absence of unfair, avoidable or remediable differences among groups of people, whether those groups are defined socially, economically, demographically, or geographically or by other dimensions of inequality (e.g. sex, gender, ethnicity, disability, or sexual orientation)”.

This reviewer feels that the third paragraph is speaking to the lack of VR programming for certain groups and that there is a lack of representation of certain groups in the programs themselves. If this interpretation is correct, this reviewer suggests the addition of a couple of sentences describing why this lack of representation is worrisome. For instance, this lack does not adequately support those with other identities but also renders these groups as being invisible and this absence itself may be an invalidating experience.

Objectives:

This reviewer is wondering if access can be defined or described. Does this refer to accessibility or usage?

Positionality statement:

This reviewer values the addition of this statement here; their positionality statement enables readers to appreciate the lens through which the authors have planned, designed, undertaken their search and analyses, and generated their discussion and conclusions.

Inclusion/exclusion criteria:

Please note that in Inclusion Criteria the SDGs are attributed to the WHO and in the Exclusion Criteria, are attributed to the UN.

Results:

“Marginalized Youth (k=3)

Multi-session, classroom-based programs provided minority students with guidance from avatars or their teachers on self-efficacy and competence (Bell et al. 2018; Wang et al. 2023).” Please define “minority students” and please consider the following:

Minority means “less than” and is now considered pejorative. In addition, groups have been made minorities by dominant culture and whiteness, thus minoritized. Importantly, marginalization and minoritization occurs not just with racial identities, but with other identities as well, including gender. At stake is the connection of status to power differentials. Minoritization is associated with a loss of power. (page 13, American Medical Association and the Association of American Medical Colleges (AAMC) Center for Health Justice “Advancing Health Equity: A Guide to Language, Narrative and Concepts”)

Underrepresented Sexualities and Genders Minorities (k=8)

This reviewer feels strongly about including the program for young female sexual violence survivors in this grouping and thinks it deserves its own “theme category,” as the one study for “immigrants”.

Intimate partner violence (IPV) is experienced by all genders and is a gendered experience as most perpetrators are men. It is a common experience: studies suggest that at least 40% of women and girls experience IPV during their lifetime. “Lumping” this study in this grouping minimizes the experiences of survivors and the power dynamic/imbalances and the cultural norms surrounding gender and relationships. It also does not fit with the other studies that look at “navigating social spaces” or “safer sexual practices.”

Putting Findings Together:

“Different emotions motivated VR use, such as feeling safe, being curious about the technology, wanting to meeting others with similar backgrounds, and being interested in experiencing new things.”

Discussion:

“Many of the studies included populations that were intersectionally marginalized, and the support programs ?had diverse aims/goals?, such as social support and affirmation, mental and emotional wellbeing, education and support for disability and physical health, and cultural connections.”

“Our findings showed that even programs unguided by a psychotherapy manual leveraged naturalistic benefits and had successful engagement.” Do programs focussed on wellbeing often come with a psychotherapy manual?

Mental and physical programming challenges and opportunities:

“Thus, replacing mainstream norms with a community-led response to wellbeing.” Sentence fragment.

“There are also opportunities to create spaces [suggest pluralization] for cultural practices (cite) and increase access to care through culturally tailored or translated triage and psychoeducation programs.”

Equity and intersectionality pitfalls and solutions:

“But, [remove comma] further study should examine the perpetuation of digital divides by VR, privileging higher-income, English-speaking communities.”

I would suggest that inequity can, in fact, be amplified by VR and other technologies, through uneven access and biased representation.

“Cultural norms are often imposed through healthcare, and thus some knowledge is discounted.” Imposed through or IN healthcare?

Recommendations:

This reviewer suggests that the authors re-consider replacing “can” with “should” or “will” in their recommendations.

“For example, prioritizing underserved populations, asking if the population has basic access needs, correcting epistemic exclusion through prioritizing indigenous healing paradigms, and other strategies to transform health systems with digital technology”: please re-write to improve clarity and capitalize “Indigenous”.

---

## [Editor Report]

Thank you for submitting your revised manuscript. The reviewer has identified several issues and provided useful recommendations. We encourage you to consider these suggestions and submit a further revised manuscript for evaluation.

---

## [Reviewer Report]

GMH-2025-0009.R3

The authors are congratulated on their scholarly work and their openness to feedback. This reviewer feels that this manuscript requires a few small edits and should be accepted for publication.

ABSTRACT:

The opening sentence of the Abstract seems to require deleting a word (“wellbeing”) to make grammatical sense and a semi-colon probably can be replaced by a comma:

Although virtual reality (VR) programs are being developed by marginalized groups, a systemic power imbalance still exists.

“Marginalized groups have a place in digital wellbeing,” might be misinterpreted to mean that marginalized groups should feel good in virtual environments, but this reviewer feels that this is not quite the intended messaging from the authors.

Pluralization of nouns such as “support” would likely be more in keeping with the intersectional lens held by the authors (e.g., “VR offered diverse support, including social, mental, physical, and cultural.”)

IMPACT STATEMENT:

Instead of “[t]he variety of uses of VR, and the fact that it can be customized, means it has the potential to be further utilized by marginalized communities,” this reviewer suggests some changes to improve clarity and emphasis: “Because of its versatility and customizability, VR holds potential for expanding its use beyond its current applications by marginalized communities.”

This reviewer suggests the first question should be split into two:

• How can VR programs be owned by marginalized populations?

• How can VR programs be sustained and grown beyond the scope and lifespan of research projects?

INTRODUCTION

Background:

First paragraph: a definition of equity should be included.

Second paragraph:

With the removal of “or,” the sentence requires further editing: “[w]hen people are excluded from resources and exposed to wellbeing risks, marginalized, they are being oppressed by multiple systems working together.”

This reviewer feels that the second paragraph would benefit from further editing. For example, “[i]n this text we use the verbs marginalized and marginalization to refer to the unfair conditions imposed on people”; please note “marginalized” is an adjective and “marginalization” is a noun. And defining “marginalization” should include additional words besides “marginalization.” The paragraph would benefit from further clarity around how “unfair conditions” relate to “marginalization” and how various “inequities” sustain this dynamic. Equity should be clearly defined in this paragraph.

Third paragraph:

“According to a review by Schueller et al. (2019) in the United States, VR development

faces access and participation barriers. These barriers are illustrated by a lack of access to technology and a lack of intersectionality and variety in programming. For example, Indigenous participants from nations across multiple high-income, Western countries indicated that digital health do not consider their needs relating to their age, gender, culture, and norms.” This reviewer is wondering how this example relates to access. The example seems to speak to: 1) lack of variety of programming; and possibly 2) lack of intersectionality. If the example is included as an example of the latter, more explanatory text would be of benefit to the reader.

This reviewer suggests reordering the sentences in the latter part of paragraph three: These issues create a feeling amongst users of lack of trustworthiness, safety, and irrelevance (Pendse et al. 2022; Whitehead et al. 2023) and can result in presenting marginalized groups a lower standard of care, amplifying healthcare inequities (Whitehead et al. 2023). Thus, the experiences within and across these marginalized groups are varied, making clear that intersectionality becomes a key consideration for VR creators.

METHODS:

Search Strategy:

“These keywords were generated from edited, pre-existing literature filters that we edited to fit our topic”: this reviewer would like to have more detail on this process.

Screening:

Inclusion criteria

Second sentence is a sentence fragment: “For example, socioeconomic, racial minorities, underrepresented sexualities or genders and persons at the intersections of these groupings.”

Exclusion criteria

UN Sustainable Development Goals should be capitalized.

It would be helpful for readers to have a second example: “[f]or example, VR aimed at improving symptoms of anxiety in individuals with anxiety disorders would be excluded,” and authors could offer another, contrasting, example of an article that would be included.

RESULTS:

Narrative Summary

Underrepresented Sexualities and Genders Minorities (k=8)

This reviewer feels strongly about including the program for young female sexual violence survivors in this grouping and thinks it deserves its own “theme category,” as the one study for “immigrants”.

This reviewer feels that this “lumping” together of two subthemes who imperfectly overlap is unfortunately artificial and somehow deemphasizes the importance of both. This reviewer also questions that if an intersectional lens is being applied to this scholarly activity, these results should not be amalgamated, but rather, given their own respective spaces.

DISCUSSION:

Leveraging naturalistic environments

Our findings showed that even programs that which were not based on psychotherapy techniques like cognitive-behavioural therapy could still leverage naturalistic benefits and had achieve successful engagement.

---

## [Editor Report]

Thank you for submitting your revised manuscript. The reviewer has identified a few minor issues and offered constructive recommendations. We are pleased to accept the manuscript on the condition that these recommendations are addressed in the final version